# Parametric Study on the Applicability of AASHTO LRFD for Simply Supported Reinforced Concrete Skewed Slab Bridges

Lucía Moya [1] and Eva O. L. Lantsoght [1,2,*]

1  Ingeniería Civil, Colegio Politécnico, Universidad San Francisco de Quito, Quito 170157, Ecuador; lmoyat@estud.usfq.edu.ec
2  Engineering Structures, Civil Engineering and Geosciences, Delft University of Technology, 2628 CN Delft, The Netherlands
*  Correspondence: e.o.l.lantsoght@tudelft.nl

**Abstract:** Simplified code provisions can be used for the analysis and design of straight slab bridges. However, several studies question the appropriateness of simplified procedures for skewed geometries. This paper provides practical insights to the designer regarding the effects of skewness in reinforced concrete slab bridges by evaluating how simplified and more refined analysis procedures impact the design magnitudes and resulting reinforcement layouts. The methods used for this study are analytical and numerical case studies. Eighty case study slab bridges with varying lengths, widths, and skew angles are subjected to the AASHTO HL-93 loading. Then, the governing moments and shear forces are determined using the AASHTO LRFD simplified procedures with hand calculations, and using linear finite element analysis (LFEA). Afterwards, the reinforcement is designed according to the AASHTO LRFD design provisions. From these case studies, it is found through the LFEA that increasing skew angles result in decreasing amounts of longitudinal reinforcement and increasing amounts of transverse flexural reinforcement. Comparing the reinforcement layouts using AASHTO LRFD-based hand calculations and LFEA, we find that using LFEA reduces the total weight of steel reinforcement needed. Moreover, as the skew increases, LFEA captures increased shear forces at the obtuse corner that AASHTO LRFD does not. In conclusion, it is preferable to design the reinforcement of skewed reinforced concrete slab bridges using LFEA instead of hand calculations based on AASHTO LRFD for cost reduction and safety in terms of shear resistance in the obtuse corners.

**Keywords:** AASHTO LRFD simplified procedures; linear finite element analysis (LFEA); live load distribution; main longitudinal reinforcement; reinforced concrete; secondary transverse reinforcement; shear reinforcement; skew angle; slab bridges

## 1. Introduction

A decisive factor when selecting a bridge type is the required span distance. Reinforced concrete slab bridges are chosen for short spans because avoiding girders can reduce labor and formwork costs [1]. Despite their limitation in span length, slab bridges are widely used. For example, in the U.S. 2020 National Bridge Inventory, nearly 10.5% of all highway bridges are classified as concrete slab bridges [2].

Slab bridges can be straight or skewed. In straight slab bridges, the main longitudinal direction of the bridge is perpendicular to the support line. In skewed slab bridges, there is a deviation of the main longitudinal direction away from the vertical axis (see Figure 1). In skewed slab bridges, the force flow is significantly more complex than in straight slab bridges [3]. However, skewed slab bridges are common when urban or geographical constraints prevent the design of straight geometries. Actually, the number of skewed bridges is growing in developing and urban cities, and the number of cases of slab bridges with a skew angle of more than 45 degrees is increasing as well [4].



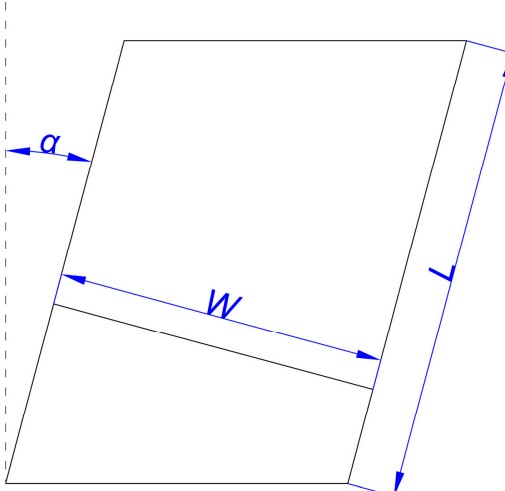

**Figure 1.** Geometric definition of length, width, and skew angle.

Numerous design codes cover the design of slab bridges [5–8]. In this work, the focus lies on the AASHTO LRFD Bridge Design Specifications (AASHTO LRFD). AASHTO LRFD is employed for design, evaluation, and rehabilitation of highway bridges [5]. The safety philosophy of this American standard is Load and Resistance Factor Design (LRFD), which is a reliability-based methodology that uses statistics to determine the appropriate safety factors for loads and resistance of components [5]. AASHTO LRFD allows the design of simply supported solid slab bridges with main longitudinal reinforcement parallel to the direction of the traffic using simplified procedures [5]. AASHTO LRFD does not prescribe the maximum skew angle for which the simplified design procedures can be applied [5].

Limited attention has been paid to skewed solid slab bridges. They were simply treated as one-way slabs where the main longitudinal moments are carried by the longitudinal reinforcement and the transverse moments are handled with empirical expressions [9]. In 2006, the collapse of the Concorde Overpass [10,11] resulted in concerns with regard to the capacity of existing reinforced concrete slab bridges. The event caused five fatal casualties and the injury of six people. This failure drew attention to the shear strength of skewed solid concrete slab bridges [12]. In the same decade, the shear capacity of existing reinforced concrete slab bridges was questioned in the Netherlands [13]. Adopting the Eurocodes [14–16] resulted in higher sectional shear forces and lower shear capacities than those used in the Dutch national codes [17,18], so that assessment of these bridges became a priority.

In recent years, efforts have been geared towards determining the effects of skew on slab bridges. This paper provides practical and relevant insights on the effects of skewness to the designer using AASHTO LRFD. More specifically, the aim is to answer the research question: How does skew influence the amount of reinforcement and its layout in reinforced concrete skewed slab bridges? To do so, the selected method is a parametric study, where AASHTO LRFD simplified procedures are compared to more refined linear finite element analyses (LFEA). This parametric study results in the main longitudinal and transverse bending moments, as well as shear forces at the obtuse corner using both approaches. These design magnitudes are then translated into reinforcement layouts. Comparing the reinforcement layouts from both methods, we can identify when larger, equal, or smaller amounts of reinforcement are found using the AASHTO LRFD hand calculations as compared to LFEA. In addition, the reinforcement layout is further translated into total steel weight to evaluate the cost. In parallel, the moment distribution capacity of the slab bridges is assessed. This is performed to determine when the same spacings or bar diameters for the main longitudinal reinforcement can be provided over the entire width of the bridge, which enhances ease of construction. As such, this work produces a tangible response on how skew influences the design of reinforced concrete

skewed slab bridges, and goes a step further than research from the literature, which focused on the design moments and shear forces.

This paper is divided in two main sections following the literature review. First, the case study bridges are described as well as the two methodologies used for analysis (AASHTO LRFD simplified procedures and LFEA) and their application towards design. Second, the results of the parametric study are presented. These are condensed within six subsections: main longitudinal bending moment, distribution width, secondary transverse bending moment, shear, influence of materials, and reinforcement comparisons based on weight of steel.

## 2. Literature Review

Several methods for understanding the effect of skew are reported in the literature. One of these is through load testing of existing bridges. Davids load tested 14 bridges and compared the rating factors obtained with AASHTO LRFD and FEA. The study showed that the rating factors increased up to 37.6% for bridges with skew angles between 15° and 20° when using FEA [19]. Other load tests of slightly skewed slab bridges have shown that the procedures for rating existing reinforced concrete slab bridges using the European codes are conservative for both shear and bending moment [20,21]. Load testing of highly skewed concrete bridges is rare. However, Bagheri developed an artificial intelligence model that can predict nondimensional frequency parameters related to the vibration modes of a slab bridge. It operates in the ranges of 0° to 60° skew angles. The input parameters are span length, deck width, deck thickness, and skew angle. With the nondimensional frequency parameters, one can calculate the flexural rigidity. This magnitude is used in the load rating and nondestructive evaluation of existing bridges; thus, the neural net is useful where structural information is incomplete [22].

Skewed slab bridges can also be studied through computational models. For example, nonlinear analysis has been used in the past [23] and in recent years [24] to study the behavior of skewed slab bridges at the ultimate limit state. Cope [23] determined that the first load that generates cracking drives the response of the slab, so that nonlinear analysis can only yield approximations of the slab's actual behavior. Hassan [24] also studied cracking load and observed that for skew angles up to 30°, the cracking load remained the same as for straight bridges, but there was a decrease when the skew reached 45°. Additionally, computational approaches have also been combined with probabilistic approaches to determine the seismic fragility of various types of skewed bridges [25].

Experimental work on skewed slab bridges is limited. Laboratory testing dating back to the eighties focused on the effect of shear in reinforced concrete slabs. One of these studies was conducted at the University of Liverpool and considered specimens with skew angles ranging from 30° to 60°. One of the main objectives of the study was to determine how to predict shear forces and evaluate shear capacity of skewed slabs. The study showed that Mindlin plate theory, with appropriate mesh refinements, can predict skewed slab behavior to a certain extent. The experiments showed that the failure mode changes from flexure to shear, and then to punching as the skew angle increases [26]. Another study, with a much more limited scope, tested two 50° skew angle scaled bridge models. The failure mode for the first specimen was flexure, and that of the second specimen, which had increased flexural reinforcement, was punching shear. Additionally, this study determined that thick plate theory could predict the initial distribution of shear stress at the obtuse corner [27]. More recently, Sharma developed a theoretical formulation to predict the ultimate flexural strength of skewed slab bridges. The outcome of the formulation was compared to results obtained from scaled test specimens with skew angles from 15° to 60°, and yielded accurate results [28].

Parametric studies provide an additional way of comprehending the response of skewed slab bridges. Some parametric studies have focused on the development of skew factors. For example, Théoret conducted a parametric numerical study on 390 simply supported slabs. This study resulted in a series of expressions for moment reduction factors

and shear magnification factors as a function of the skew angle. These factors compensate for skewness when using the simplified analysis procedures from AASHTO LRFD [9]. Similarly, skew factors that increment load effects were developed in the Netherlands for bridge assessment [29].

Other parametric numerical studies have focused on force distribution and concentration in skewed slab bridges. Menassa analyzed 96 case study bridges using the AASHTO Standard Specifications, AASHTO LRFD, and LFEA. The research, which focused on bending moments, confirmed that skewed slab bridges can be designed as straight for skew angles smaller than 20° [30]. Likewise, Hulsebosch developed a parametric study with a focus on the influence of skew towards the magnitudes of bending moments and shear forces. He determined that the addition of ATS (additional triangular segments) adjacent to the free edges of the slab bridge reduces the governing shear forces at the obtuse corners [12], and recommended this practice for the design of new skewed slab bridges. Additionally, Fawaz analyzed 96 case study bridges with a special attention on the influence of railings on bending moments. The parametric study showed that the presence of railings, on top of the skew angle, can further reduce the main longitudinal bending moments obtained with AASHTO LRFD in skewed slab bridges [31].

From the literature review, we identified the research gap as a parametric study on the resulting reinforcement layout in reinforced concrete skewed slab bridges. By focusing on the resulting reinforcement, this paper provides the designer with practical insights. Additionally, since there is no clear consensus on how to evaluate the shear capacity of slab bridges [14], the application of a new approach is presented herein. The selected procedure comes from Lipari, who proposed variations to extend shear design code provisions for straight geometries to skewed geometries [32]. These procedures will be elaborated in Section 3.2.2.

## 3. Materials and Methods

### 3.1. Design of the Parameter Studies

The three geometric parameters studied are length (*L*), width (*W*), and skew angle (*α*). The length is taken as the dimension of the free edge, the width is considered as the dimension perpendicular to the free edge, the skew angle is measured as the angle between the vertical axis (dotted line) and the free edge (see Figure 1), and the driving direction is parallel to the free edge.

Additionally, the cross-section includes the design lane(s), one shoulder and one concrete barrier at each edge, and a 50 mm thick future wearing surface covering the design lane(s) and shoulders. The cross-section is kept the same for all case study bridges, and only the number of lanes is varied. Figure 2 shows the layout for the case study bridge with one lane. Case study bridges with more lanes follow a similar layout.

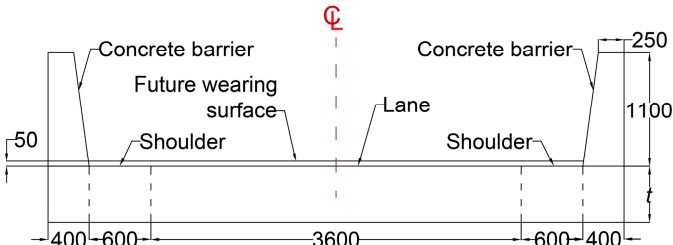

**Figure 2.** Cross-section for case study bridge with one lane. All dimensions are in millimeters.

To design the reinforcement, we used 500 MPa steel and concrete with a compressive strength of 35 MPa. The length, *L*, is varied from 7.5 to 15 m in increments of 2.5 m, representing a typical range of lengths used for solid slab bridges. The width is varied from 5.6 (one lane) to 16.4 m (four lanes) in increments of 3.6 m (one design lane). Finally, the skew angle is varied from 0° to 60° in increments of 15°, representing a typical range of skew angles used for skewed solid slab bridges. In the Netherlands, skew slab bridges of approximately 15°, 30°, 45°, and 60° skew angles represent roughly 56%, 26%, 14%, and 4%

respectively, of the total skewed slab bridges registered in the country [33]. Every possible parameter combination is modeled, resulting in 80 different case study bridges.

Additionally, the parametric study is extended by studying the influence of the material parameters using two case study bridges as a reference. The yield strength of the steel is taken as 500 and 220 MPa, and the concrete compressive strength is taken as 25, 35, and 60 MPa. The lower yield strength of the steel represents the steel grade that is found in existing slab bridges. The lower concrete compressive strength represents both a deteriorated concrete as well as a regular low-strength concrete. However, existing slab bridges often have a higher concrete compressive strength as a result of the ongoing hydration of the cement, which is represented in this case study by using the 60 MPa concrete [34,35]. This higher strength is also explored for the design of new reinforced concrete slab bridges with high skew angles [12]. The first reference case study bridge that is used for evaluating the effect of material parameters has a 7.5 m span, 4 lanes, and a 15° skew. The second one is the 15 m span, 2 lanes, and 45° skew case study bridge.

The resulting total number of case study bridges is 90. All parameters are summarized in Table 1. The 80 configurations of geometry are illustrated in Figure 3. All configurations are analyzed with the AASHTO LRFD simplified procedures, using hand calculations, and with LFEA. The governing load combination comes from the AASHTO LRFD for the limit state Strength I.

**Table 1.** Overview of parameters studied.

| $L$ (m) | Number of Lanes | $W$ (m) | $\alpha$ (°) | $f_y$ (MPa) | $F'_c$ (MPa) |
|---|---|---|---|---|---|
| 7.5 | 1 | 5.6 | 0 | 500 | 25 |
| 10 | 2 | 9.2 | 15 | 220 | 35 |
| 12.5 | 3 | 12.8 | 30 | - | 60 |
| 15 | 4 | 16.4 | 45 | - | - |
| - | - | - | 60 | - | - |

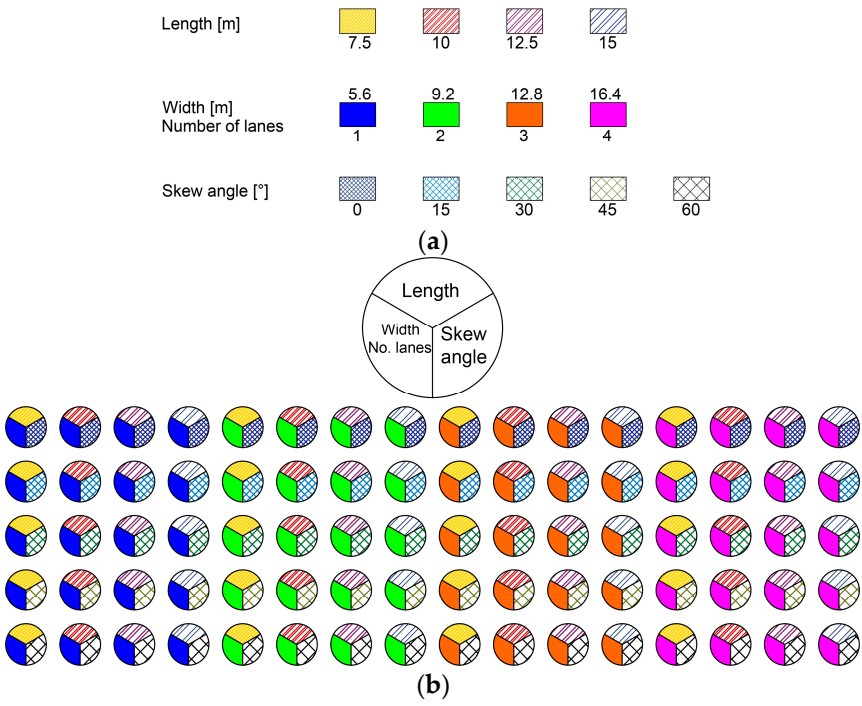

**Figure 3.** Illustration of the 80 case study bridges. (**a**) Parameters considered, and (**b**) geometric properties of the case study bridges.



*3.2. Analysis and Design Procedures*

3.2.1. AASHTO LRFD with Hand Calculations

The simplified method from AASHTO LRFD allows single-span, reinforced concrete, solid slab bridges to be analyzed as a number of simply supported beams [5]. The following loads are considered:

- DC is the self-weight of the slab bridge and the concrete barriers, using a unitary weight of 24.52 kN/m$^3$. The self-weight of the slab is uniformly distributed over the whole surface. The self-weight of the concrete barriers acts on the exterior strips [36].
- DW is the self-weight of the future wearing surface, using a unitary weight of 22.78 kN/m$^3$. The self-weight is uniformly distributed over the design lanes and shoulders.
- LL uses the AASHTO HL-93 combination for the vehicular live load. The lane load is uniformly distributed over a 3.05 m (10 ft) width. The design truck or tandem is applied as point loads so that it generates the most critical moment or shear, depending on the considered failure mode. A dynamic load allowance is considered as well [5].

The simply supported strips are analyzed as beams. The self-weight from DC and DW is applied as a uniformly distributed line load. The procedures for calculating the equivalent strip widths for slab bridges are as follows:

- The equivalent width for interior strips considering one loaded lane is taken as [5]:

$$E = 10.0 + 5.0\sqrt{L_1 W_1} \qquad\qquad E(in); \ L_1(ft); \ W_1(ft) \qquad\qquad (1)$$

  where $E$ is the equivalent width, $L_1$ is the span of the bridge taken as the lesser of the real span and 18.29 m (60 ft), and $W_1$ is the width of the bridge taken as the lesser of the real width and 9.14 m (30 ft).
- The equivalent width for interior strips considering multiple loaded lanes is taken as [5]:

$$E = 84.0 + 1.44\sqrt{L_1 W_1} \leq \frac{12.0W}{N_L} \qquad\qquad E(in); L_1(ft); W_1(ft); W(ft) \qquad (2)$$

  where $E$ is the equivalent width, $L_1$ is the span of the bridge taken as the lesser of the real span and 18.29 m (60 ft), $W_1$ is the width of the bridge in feet taken as the lesser of the real width and 18.29 m (60 ft), $W$ is the total width of the bridge taken from edge-to-edge, and $N_L$ is the number of design lanes.
- The reduction factor applied to interior strips of skewed bridges is taken as [5]:

$$r = 1.05 - 0.25\tan(\alpha) \leq 1.00 \qquad\qquad (3)$$

  where the angle $\alpha$ is the skew angle in degrees. The reduction factor is then multiplied by the equivalent strip widths calculated with (1) and (2). The larger result is taken as the interior equivalent strip width. Afterwards, the maximum design moment or shear for live load is divided by the equivalent strip width, resulting in the design live moment or shear for the interior strip [5].
- The equivalent width for exterior strips is taken as the distance between the inside face of the barrier and the edge of the deck, plus 305 mm (12.0 in), and plus a quarter of the strip width. This value should not exceed 1829 mm (72.0 in) or half the full interior strip width. Then, the maximum design moment or shear for live load is obtained by considering one line of wheels from the vehicle in the HL-93 load model and a tributary portion of the design lane load. Afterwards, this magnitude is divided by the equivalent strip width, resulting in the design live load moment or shear for the exterior strip [5].

The midspan moment for the DC, DW, tandem, and lane load is used for the flexural design. For the truck, the maximum moment follows from positioning the vehicle so that

the midspan point of the bridge bisects the distance between the nearest 142 kN (32 k) axle and the center of gravity of the vehicle [2].

For shear design, the moments and shears are taken at the critical shear section. This location is measured perpendicularly at a distance, $d_v$, from the support line (see Figure 4a for reference). The distance, $d_v$, is the effective shear depth, which is the distance between the resultant of the compressive and the resultant of the tensile forces, not exceeding 0.72 $h$, with $h$ being the height of the cross-section [5]. The heaviest axle of the vehicle is positioned at the location of the critical shear section.

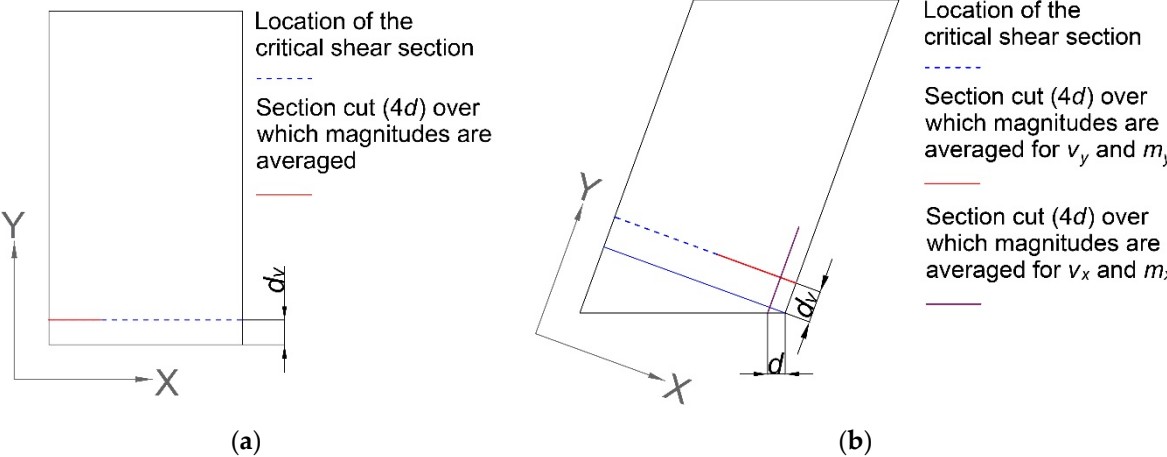

**Figure 4.** Shear section cuts for design. (**a**) Straight bridges and (**b**) skewed bridges.

With these design magnitudes, the main longitudinal reinforcement is designed for the Strength I limit state. The secondary transverse reinforcement is calculated with the distribution steel provisions from AASHTO LRFD [5]. Finally, the need for shear reinforcement is verified with the design procedures from AASHTO LRFD [5]. The shear provisions are a simplified version of the Modified Compression Field Theory [37]. Detailed calculations for each of the case study bridges can be found in the open access PDF document titled Calculation Memories provided as part of the Supplementary Materials.

### 3.2.2. Use of Linear Finite Element Models

LFEA is performed using SCIA Engineer version 20.0 [38]. The solid slab bridges are modeled with isotropic shell elements. The element size is 100 mm. While the majority of FEM software offer Mindlin and Kirchhoff plate bending theory for the analysis, Mindlin theory is chosen as suggested by Hulsebosch [12]. The slabs are supported on hinged line supports on the sides adjacent to the free edges. Additionally, the loads are as described for the simplified methods with a few modifications. First, the vehicle loading is applied as uniformly distributed loads acting over the standard tire contact area [5]. Second, the multiple presence factor $m$, which multiplies the magnitude obtained for the live loading, is included (see Table 2) [5]. This multiple presence factor represents the limited probability of having multiple lanes loaded simultaneously throughout the design life of the bridge. Third, the barrier is modeled as a uniformly distributed load located along the base width of the barrier on both edges (see Figure 2).

**Table 2.** Multiple presence factors [5].

| Number of Loaded Lanes | Multiple Presence Factors |
| :---: | :---: |
| 1 | 1.2 |
| 2 | 1 |
| 3 | 0.85 |
| >3 | 0.65 |

The design longitudinal bending moment for DC and DW is taken as the peak value of the section cut performed perpendicular to the longitudinal direction of the bridge. This section cut is done at the location of the maximum longitudinal bending moment (see Figure 4). The design bending moment for LL is averaged over the effective width (see Figure 5). The distance where the resisting action caused by the maximum stress is distributed along the effective width is the same as the resisting action caused by the variable stresses along the entire width [39,40]. The live load distribution width is obtained as the weighted average between the distribution width and maximum moment for the lane and vehicle loading cases.

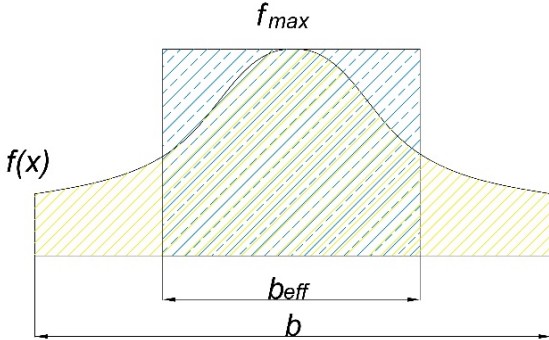

**Figure 5.** Principle of the effective width applied to moment determination [41].

A section cut parallel to the longitudinal direction is used to determine the design moments for transverse flexural reinforcement. This cut is made at the location of the maximum secondary transverse bending moment (see Figure 6). Then, the bending moment is obtained by averaging the Strength I load combination over the effective width.

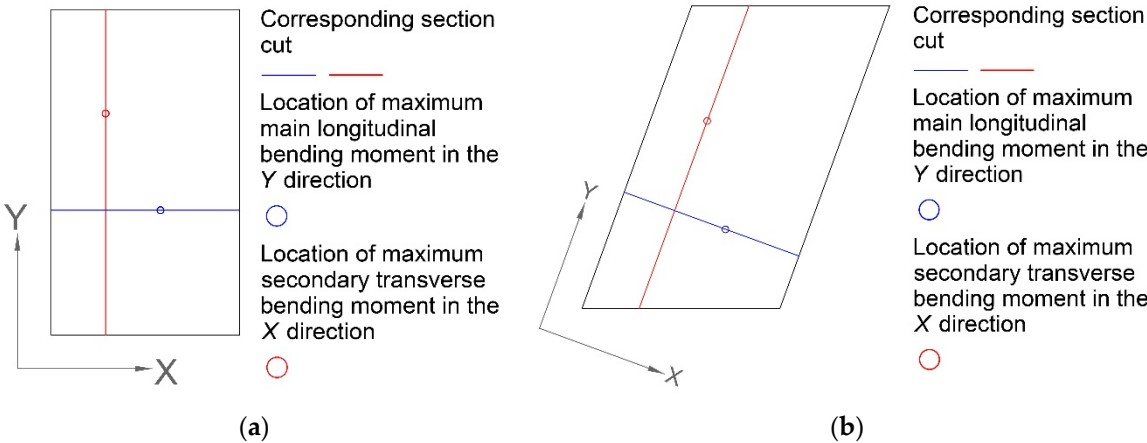

**Figure 6.** Main longitudinal and transverse moment section cuts for design. (**a**) Straight bridges and (**b**) skewed bridges.

The design moments and shear forces for shear design are taken at the location of the critical shear section. They are resolved in the direction of the principal shear force, which is calculated as follows [32]:

$$v_0 = \sqrt{v_x^2 + v_y^2} \tag{4}$$

Then, the direction of the principal shear force, which is expected to be comparable to the skew angle, is given by [32]:

$$\theta_0 = \arctan\left(\frac{v_y}{v_x}\right) \tag{5}$$

Finally, the moment in the direction of the principal shear force is calculated as [32]:

$$m_0 = m_x \cos^2 \theta_0 + m_y \sin^2 \theta_0 + 2m_{xy} \cos \theta_0 \sin \theta_0 \tag{6}$$

The magnitudes for shear and moment at the critical shear section are mesh-dependent. Therefore, they are averaged over a width of 4 $d$, with $d$ being the effective depth to the main longitudinal reinforcement (see Figure 4). This distribution width is determined by comparing LFEA models and shear tests on straight reinforced concrete slabs in the laboratory [42]. It is also applied in the guidelines for the assessment of existing bridges in the Netherlands [43].

Using the values obtained from processing the LFEA results, the reinforcement is designed for the Strength I limit state. Then, it is checked if the distribution steel fulfills the transverse moment demands as the skew increases. Finally, the need for minimum shear reinforcement is determined. To do so, the area of the main longitudinal reinforcement is resolved in the direction of the principal shear force as follows [32]:

$$A_{s_{calc},\alpha_x}(\theta_0) = A_{s,\alpha_x} \cos^4(\theta_0 - \alpha_x) \tag{7}$$

where the angle $\theta_0$ represents the direction of the principal shear force, and the angle $\alpha_x$ is the skew angle with respect to the $x$-axis. The nominal shear capacity is calculated twice. First, it is assumed that the main flexural reinforcement is parallel to the direction of the traffic. Then, a virtual rotation of the main flexural reinforcement is assumed. These two magnitudes are compared to validate the design procedure of taking moments and shears in the direction of the principal shear stress and using an equivalent reinforcement area [32]. Detailed calculations for each of the case study bridges can be found in the open access PDF document titled Calculation Memories provided as part of the Supplementary Materials.

## 4. Results and Analysis

### 4.1. Analysis Methodology

We first compare the maximum longitudinal bending moments from AASHTO LRFD and LFEA, as well as the resulting reinforcement using LFEA and interior strip AASHTO LRFD. Then, the distribution widths for live load from LFEA are compared to the total width of the cross-section. Subsequently, a similar comparison is made for the transverse bending moments. Next, shear demand and shear capacity are compared for both AASHTO LRFD and LFEA to determine when minimum shear reinforcement is needed. Afterwards, the results are compared for the case study bridges with varied material parameters. Finally, the designed reinforcement is compared in terms of weight of steel.

All comparisons are based on the Strength I limit state. For the Service I limit state, we would need to place the reinforcement in two layers in some cases or use a higher slab depth. Changing reinforcement layout and depth would have impeded the one-on-one comparisons of the study, and thus we did not include these changes in our designs.

### 4.2. Maximum Main Longitudinal Bending Moment

Figure 7 compares the results of LFEA and AASHTO LRFD in terms of maximum longitudinal bending moment as a function of the skew angle. Figure 7a–d show the results for the different span lengths. The data points for LFEA are specific for each case study bridge. Those for AASHTO LRFD represent upper and lower bounds. AASHTO LRFD design moments are higher for exterior than interior strips. As a result, the dash-dotted line in Figure 7 for the exterior strip provides the upper bound design moments, and the dashed line for interior strip provides the lower bound.

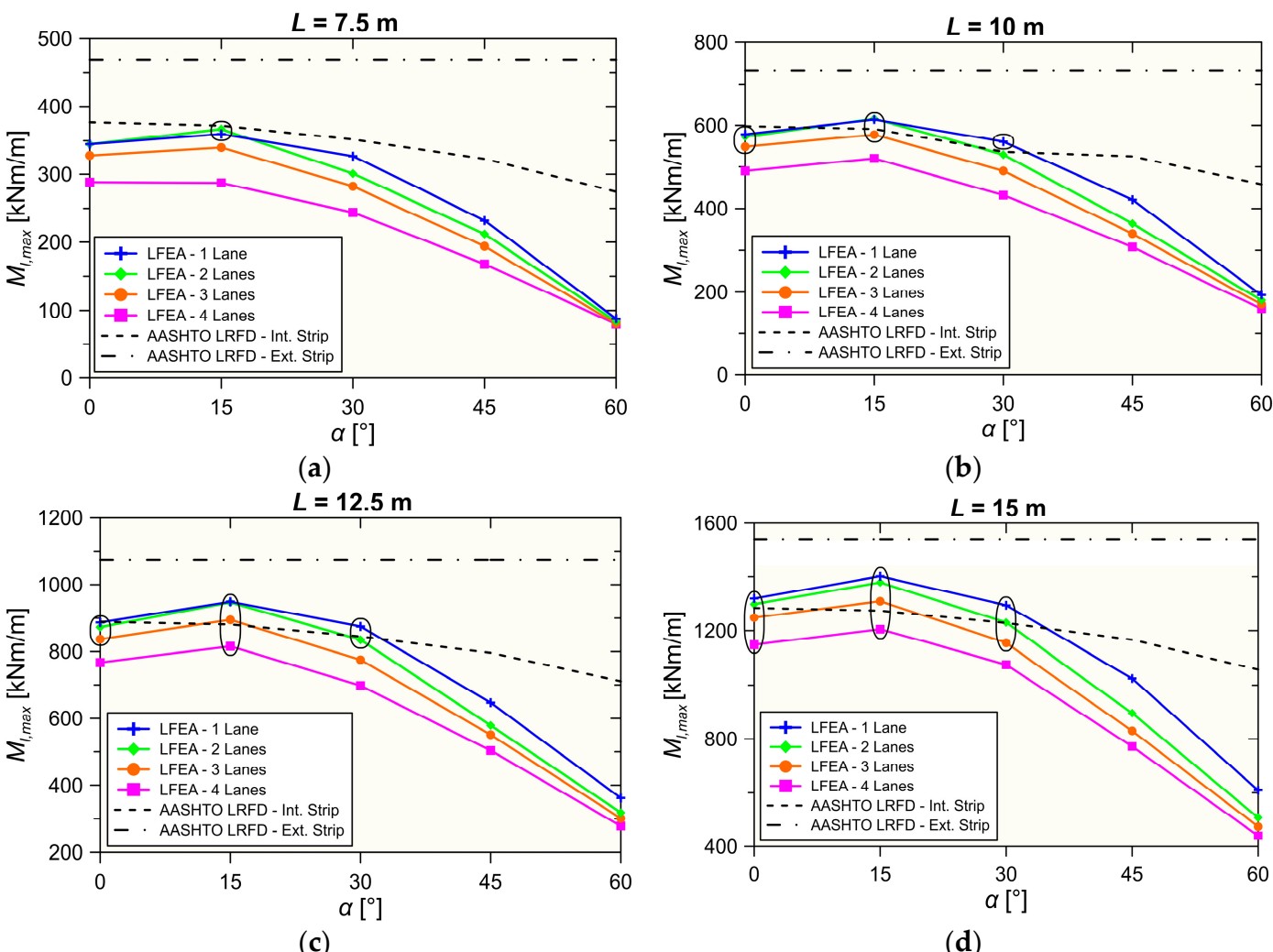

**Figure 7.** Comparison of LFEA and AASHTO LRFD maximum longitudinal bending moment. (**a**) *L* = 7.5 m, (**b**) *L* = 10 m, (**c**) *L* = 12.5 m, (**d**) *L* = 15 m. Circled data indicates that the same reinforcement was provided with LFEA and AASHTO LRFD.

For the 7.5 m long bridges (see Figure 7a), the only cases where the same reinforcement is provided using AASHTO LRFD and LFEA is for the one- and two-lane case study bridges with a 15° skew angle. For the 10 m long bridges (see Figure 7b), the same reinforcement is provided for the one-, two-, and three-lane case study bridges with a 0° and 15° skew angle. This also holds true for the one-lane case study bridge with a 30° skew angle. Likewise, for the 12.5 m long bridges, the same reinforcement is provided for the one-, two-, and three-lane straight bridges. In addition, all the case study bridges with a 15° skew as well as the one- and two-lane case study bridges with a 30° skew angle are designed with the same reinforcement using both methods. For the 15 m span bridge (Figure 7d), the same reinforcement is provided for all case study bridges with a 0° and 15° skew angle. Aside from these, the one-, two-, and three-lane case study bridges with a 30° skew angle obtain the same reinforcement using both methods.

As can be seen in Figure 7, for the majority of case study bridges, the resulting bending moment with LFEA is less than the AASHTO LRFD interior strip. AASHTO LRFD only captures a small reduction of resulting longitudinal bending moments as the skew increases. In comparison, LFEA captures a small increase between 0° and 15° and then the bending moment drops significantly as the skew increases between 15° and 60°.

Table 3 summarizes the results from Figure 7. Columns two and three show, for the straight case study bridges, the overestimation of the longitudinal bending moment using

AASHTO LRFD as compared to LFEA, expressed as a percentage. Since this percentage varies depending on the number of lanes, an average is presented in Table 3. The overestimation is larger for shorter spans, reaching 22.5% for interior strips and 39.8% for exterior strips. The overestimation for the exterior strips is significantly larger than the overestimation for interior strips.

**Table 3.** Interior and exterior strip $M_{l,max}$ overestimation and $M_{l,max}$ reduction (in percentage) using AASHTO LRFD and LFEA.

| Length (m) | Interior Strip $M_{l,max}$ Overestimation (%) | Exterior Strip $M_{l,max}$ Overestimation (%) | $M_{l,max}$ Reduction for 60° Skew per AASHTO LRFD (%) | $M_{l,max}$ Reduction for 60° Skew per LFEA (%) |
|---|---|---|---|---|
| 7.5 | 22.5 | 39.8 | 27.9 | 75.3 |
| 10 | 13.9 | 33.3 | 24.0 | 69.8 |
| 12.5 | 8.9 | 28.9 | 20.6 | 65.1 |
| 15 | 5.5 | 25.1 | 18.1 | 61.8 |

Columns four and five in Table 3 indicate the percentage decrease between the 60° case and the straight case for $M_{l,max}$, using the AASHTO LRFD method (column 4) and LFEA (column 5). For the AASHTO LRFD procedures, the results for the interior strip are considered. Since the percentage varies depending on the number of lanes, an average is presented in Table 3. As can be seen, the moment reduction that AASHTO LRFD is able to capture is nearly 40% less than that attained by LFEA for all span lengths considered.

The reduction of longitudinal bending moment that occurs as the skew angle increases is explained by the trajectories of the principal stresses. For straight bridges (see Figure 8a), the direction of the principal stresses follows the longitudinal direction of the bridge. This causes the main longitudinal bending moments to be higher. However, as the skew increases (see Figure 8b), the trajectories shift from the longitudinal to the transverse direction. The result of this change in trajectories is the reduction of the main longitudinal bending moments shown in Figure 7.

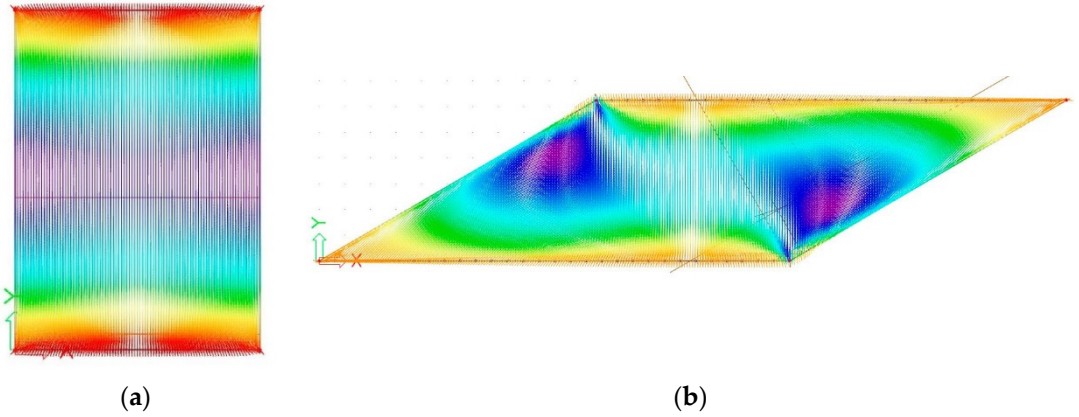

(**a**)  (**b**)

**Figure 8.** Trajectories of the principal stresses for the 12 m long, two-lane case study bridge. (a) Straight $\alpha = 0°$, (b) skewed $\alpha = 60°$.

### 4.3. LFEA Live Load Distribution Width for Main Longitudinal Bending Moment

Figure 9 shows the effective width for live load from LFEA ($W_{eff}$) to the width of the bridge ($W$). By dividing these two magnitudes, the proportion of the width of the bridge that carries live load is determined. This proportion is referred to as the width factor ($W_{fact}$) (see Equation (8)). Continuous lines in Figure 9 signal the upper bound width factor for case study bridges sharing a same number of lanes, and dashed lines signal the lower bound. The data points indicate specific case study bridges, but no distinction is made as to their specific span length. The horizontal line at $W_{fact} = 0.5$ identifies that the live load is

distributed over half the width of the bridge. As shown in Figure 9, the majority of case study bridges have an effective width larger than half of the bridge width. In fact, 68 of the 80 case study bridges carry live load width over a third of the width, 55, more than one half, and 42, more than three quarters. Additionally, Figure 9 shows that wider bridges tend to have a lower $W_{fact}$ than narrower bridges. Moreover, shorter span bridges have a lower $W_{fact}$ than longer span bridges. For instance, 30 out of the 40 case study bridges with span lengths of 12.5 and 15 m have a $W_{fact}$ of more than 50%. On the other hand, only 11 out of the 20 case study bridges with span length of 7.5 m have a $W_{fact}$ of more than 50%.

$$W_{fact} = \frac{W_{eff}}{W} \tag{8}$$

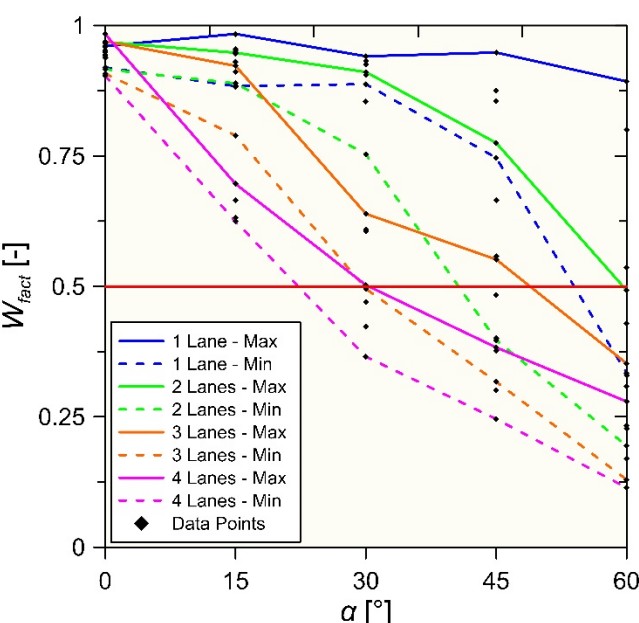

**Figure 9.** Relation of effective width to bridge width for live load bending moment from LFEA, showing data of all 80 case study bridges.

As mentioned before, when the skew angle increases, the trajectories of the principal stresses shift away from the longitudinal to the transverse direction (see Figure 8). However, this is not the only effect that the trajectories undergo. In fact, skewness also causes the trajectories of the principal stresses to concentrate towards the edges. This phenomenon results in a reduction of $W_{fact}$ for increasing skew angles. More concentrated live load moment concentrations result in a smaller $W_{fact}$.

### 4.4. Maximum Secondary Transverse Bending Moment

Figure 10 compares the results of LFEA and AASHTO LRFD in terms of maximum transverse bending moment as a function of the skew angle. Figure 10a–d show the results for the different span lengths. The data points for LFEA are specific to each case study bridge. The lines for AASHTO LRFD represent upper and lower bounds. Data from LFEA represents the maximum transverse design moment. AASHTO LRFD requires a design based on the distribution reinforcement provisions, so that $M_{tmax}$ for these cases is the capacity provided by the distribution reinforcement [5].

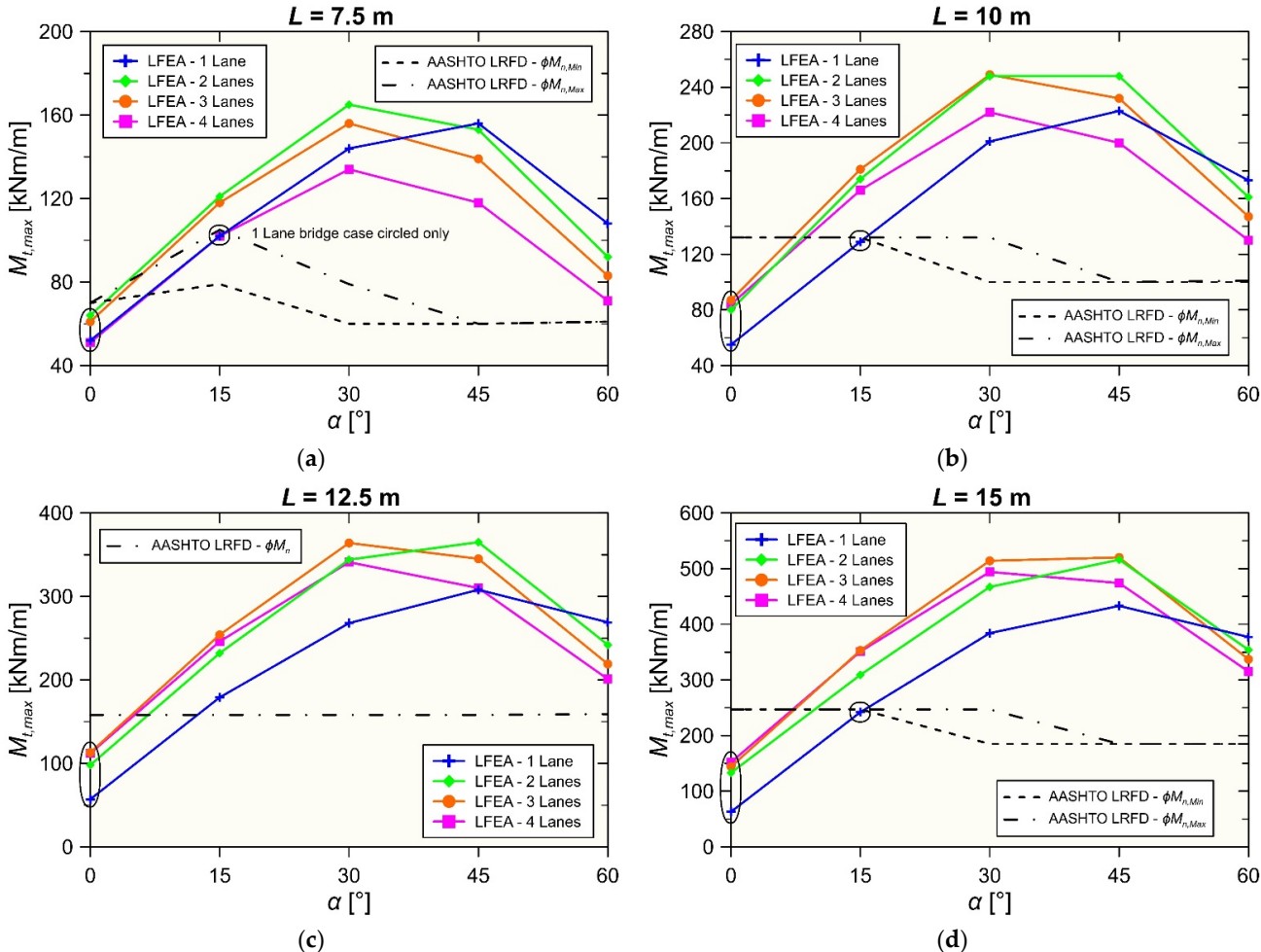

**Figure 10.** Comparison of LFEA maximum transverse bending moment and AASHTO LRFD bending moment capacity provided by distribution reinforcement. (**a**) *L* = 7.5 m, (**b**) *L* = 10 m, (**c**) *L* = 12.5 m, (**d**) *L* = 15 m. Circled data indicates that the same reinforcement was provided as for AASHTO LRFD.

For the 7.5 m span bridges (see Figure 10a), the transverse reinforcement designed with AASHTO LRFD is only able to match the demand determined with LFEA for the one-lane and 15° skew bridge, besides the straight cases. For the 10 m span bridges (see Figure 10b), the AASHTO LRFD distribution reinforcement is only sufficient for the demand determined with LFEA for the one-lane and 15° skew angle case study bridge. For the 12.5 m span bridges (see Figure 10c), the AASHTO LRFD distribution reinforcement is insufficient for the demand determined with LFEA for all skewed cases. For the 15 m span bridges (see Figure 10d), the AASHTO LRFD distribution reinforcement provisions only match the demand for the one-lane and 15° skew bridge. While the transverse bending moments with LFEA are influenced by the skew (see Figure 8), the distribution reinforcement capacity from AASHTO LRFD is not. We can conclude that the AASHTO LRFD distribution reinforcement is not sufficient for the majority of the skewed bridges.

Table 4 summarizes the results from Figure 10. Columns two through five indicate the percentage increase of the $M_{t,max}$. The increase results from comparing the demand from LFEA to the moment capacity achieved when only distribution reinforcement as per AASHTO LRFD is provided. Since the percentage varies depending on the number of lanes, an average is presented in Table 4. This increase is largest for the 30° and 45° skew angle case study bridges, where the percentage increase is around 100%. The highest percentage is 162.6% for the 15 m span length and 45° skew angle. These values show that AASHTO LRFD distribution reinforcement is not sufficient for skewed bridges.

**Table 4.** Increase in LFEA $M_{t,max}$ demand versus AASHTO LRFD capacity.

| Length (m) | 15° Skew Angle (%) | 30° Skew Angle (%) | 45° Skew Angle (%) | 60° Skew Angle (%) |
|---|---|---|---|---|
| 7.5 | 23.4 | 103.0 | 135.8 | 45.1 |
| 10 | 23.7 | 87.7 | 125.8 | 52.1 |
| 12.5 | 44.1 | 108.4 | 110.1 | 46.7 |
| 15 | 27.5 | 104.9 | 162.6 | 86.9 |

The increase of transverse bending moment for increasing skew angles is explained by the trajectories of the principal stresses. For straight bridges (see Figure 8a), the direction of the principal stresses follows the longitudinal direction of the bridge. As a result, the transverse bending moments are minimal and can be resisted by minimum secondary transverse reinforcement. When the skew grows, the trajectories shift from the longitudinal to the transverse direction (see Figure 8b). In consequence, the transverse bending moments increase and require more than minimum reinforcement.

### 4.5. Shear Demand versus Shear Capacity

Figure 11 shows the ratio (Unity Check, UC, see Equation (9)) of shear demand ($V_u$) to factored concrete shear capacity ($\phi V_c$) as a function of the skew angle. Shear reinforcement is required when UC > 1, with UC defined as:

$$\text{UC} = \frac{V_u}{\phi V_c} \tag{9}$$

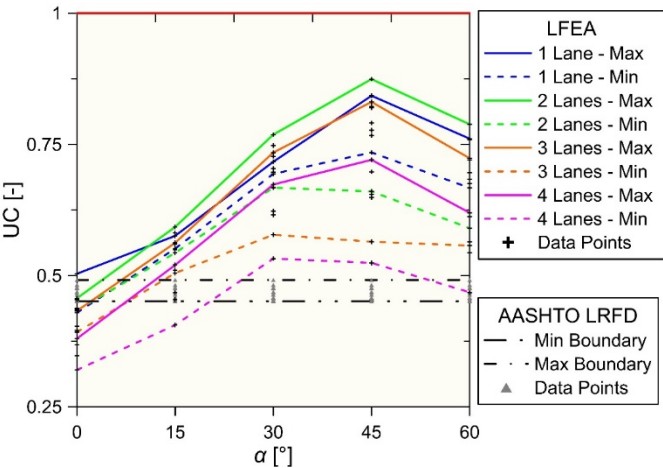

**Figure 11.** Relation of shear demand to concrete shear capacity.

Continuous lines show the upper bound values determined from LFEA for the case study bridges of the same span length, and dashed lines show the lower bound. Dash-dotted horizontal lines show the maximum and minimum boundaries for the AASHTO LRFD UC. The data points indicate specific case study bridges. The horizontal line at UC = 1 signals the point in which shear reinforcement is necessary to fulfill the AASHTO LRFD design provisions for shear [5].

As shown in Figure 11, none of the case study bridges analyzed with AASHTO LRFD or with LFEA required shear reinforcement. The highest unity check in the obtuse corner was 0.874 for the two-lane, 15 m span, and 45° skew angle case study bridge analyzed with LFEA. The relation between the UC on the skew angle is different for the AASHTO LRFD and LFEA approaches. The UCs for AASHTO LRFD are not influenced by the skew. The UCs using the demand from LFEA increase from 0° to 30°, then either increase or decrease slightly from 30° to 45°, and decrease from 45° to 60°. This behavior is explained

by the fact that skewness causes stress concentrations in the obtuse corners. These stress concentrations are translated into substantial shear peak stresses that do not occur in straight geometries. The straight case study bridges have significantly lower UCs than the skewed ones. For the skewed bridges, a reduction in the concrete compressive strength could lead to the need for shear reinforcement.

### 4.6. Influence of Material Properties

Two case study bridges were designed for the six possible combinations of 500 and 220 MPa yield strength steel and 25, 35, and 60 MPa compressive strength concrete. The aim is to determine how the materials' properties influence the reinforcement design. The case numbers for the 12 bridges from this section respond to the numbering adopted in the Calculation Memories found in the Supplementary Materials. If for each skew angle (0°, 15°, 30°, 45°, and 60°) we considered the aforementioned 6 possible material combinations, 96 case study bridges would result (see Equation (10)). Therefore, the 0° skew angle bridges were numbered from 1 through 96, the 15° skew angle bridges from 97 through 193, and so on.

$$Case\ study\ bridges\ per\ skew\ angle = 6\ [material\ combinations] \times 4\ [width\ variations] \times 4\ [length\ variations] = 96\ case\ study\ bridges \tag{10}$$

The selected base case study bridges are those with the highest and lowest shear UC for the 35 MPa concrete and 500 MPa steel. These two case study bridges are identified in bold in Tables 5 and 6. The bridge with the lowest shear UC was 7.5 m long, had 4 lanes, and a 15° skew angle. The bridge with the highest shear UC was 15 m long, had 2 lanes, and a 45° skew angle.

**Table 5.** Designed reinforcement for the 7.5 m long, four-lane, 15° skew case study bridge using LFEA when varying the material properties.

| Bridge | Material | | Longitudinal Design | | Transverse Design | | Shear Design | |
|---|---|---|---|---|---|---|---|---|
| Case | $f'_c$ (MPa) | $f_y$ (MPa) | $A_{s,calc}$ (mm$^2$/m) | Design (mm @ mm) | $A_{s,calc}$ (mm$^2$/m) | Design (mm @ mm) | UC | Reinforcement |
| 125 | 25 | 500 | 1621 | φ 25 @ 250 | 591 | φ 16 @ 300 | 0.489 | Not needed |
| 109 | 35 | 500 | 1598 | φ 25 @ 250 | 588 | φ 16 @ 300 | **0.406** | Not needed |
| 141 | 60 | 500 | 1576 | φ 25 @ 250 | 585 | φ 16 @ 300 | 0.305 | Not needed |
| 173 | 25 | 220 | 3684 | φ 25 @ 100 | 1350 | φ 20 @ 200 | 0.370 | Not needed |
| 157 | 35 | 220 | 3632 | φ 25 @ 100 | 1343 | φ 20 @ 200 | 0.308 | Not needed |
| 189 | 60 | 220 | 3581 | φ 25 @ 100 | 1336 | φ 20 @ 200 | 0.233 | Not needed |

**Table 6.** Designed reinforcement for the 15 m long, two-lane, 45° skew case study bridge using LFEA when varying the material properties.

| Bridge | Material | | Longitudinal Design | | Transverse Design | | Shear Design | |
|---|---|---|---|---|---|---|---|---|
| Case | $F'_c$ (MPa) | $f_y$ (MPa) | $A_{s,calc}$ (mm$^2$/m) | Design (mm @ mm) | $A_{s,calc}$ (mm$^2$/m) | Design (mm @ mm) | UC | Reinforcement (mm @ mm) |
| 312 | 25 | 500 | 2947 | φ 28 @ 200 | 1721 | φ 20 @ 150 | 1.054 | 2 φ 10 @ 15 |
| 296 | 35 | 500 | 2903 | φ 28 @ 200 | 1706 | φ 20 @ 150 | **0.874** | Not needed |
| 328 | 60 | 500 | 2860 | φ 28 @ 200 | 1691 | φ 20 @ 150 | 0.658 | Not needed |
| 360 | 25 | 220 | 6719 | φ 32 @ 100 | 3952 | φ 25 @ 100 | 0.696 | Not needed |
| 344 | 35 | 220 | 6618 | φ 32 @ 100 | 3916 | φ 25 @ 100 | 0.578 | Not needed |
| 376 | 60 | 220 | 6519 | φ 32 @ 100 | 3881 | φ 25 @ 100 | 0.435 | Not needed |

For cases with the same yield strength of the steel, the longitudinal and transverse reinforcement is not influenced by the concrete compressive strength (see Tables 5 and 6). The concrete compressive strength barely influences the theoretical area of required steel, $A_{s,calc}$. The difference in $A_{s,calc}$ between 25 and 60 MPa concrete is 3%. This observation

is explained because the concrete compressive strength has only a small influence on the height of the compressive stress block. As a result, the influence on the internal lever arm at the ultimate limit state and flexural capacity is minimal. The shear unity check varies significantly as the concrete compressive and steel yield strength change. This variation can lead to the need for shear reinforcement, as can be in Table 6 for the UC marked in red. The results in Tables 5 and 6 show that lower concrete compressive strengths and higher steel yield strengths lead to higher unity checks. This observation can be explained by the square root relationship between concrete compressive strength and concrete shear capacity. A section that uses a higher yield strength steel requires less area of steel for flexural reinforcement. The net longitudinal tensile strains thus increase, reducing the parameter $\beta$ used for determining the concrete shear capacity [5]. $\beta$ is a shear parameter that is a function of axial strain and level of applied shear stress [5].

### 4.7. Weight of Steel Reinforcement Comparison

This section provides a comparison between the steel needed for the flexural reinforcement of the 80 case study bridges designed with AASHTO LRFD and LFEA. The weight calculation assumes that the same bar diameters and spacings are provided along the whole width and length of the bridge for the longitudinal and transverse reinforcement. In other words, for the longitudinal reinforcement, the comparison is based on the interior strip only. Sixteen representative bridges are selected to illustrate the observations, and their properties are given in Table 7.

**Table 7.** Description of bridge identifiers for Figures 12 and 13 and Tables 8–10.

| Bridge Identifier | Description | Bridge Identifier | Description |
|---|---|---|---|
| 1 | 7.5 m span, 1 lane | 9 | 7.5 m span, 3 lanes |
| 2 | 10 m span, 1 lane | 10 | 10 m span, 3 lanes |
| 3 | 12.5 m span, 1 lane | 11 | 12.5 m span, 3 lanes |
| 4 | 15 m span, 1 lane | 12 | 15 m span, 3 lanes |
| 5 | 7.5 m span, 2 lanes | 13 | 7.5 m span, 4 lanes |
| 6 | 10 m span, 2 lanes | 14 | 10 m span, 4 lanes |
| 7 | 12.5 m span, 2 lanes | 15 | 12.5 m span, 4 lanes |
| 8 | 15 m span, 2 lanes | 16 | 15 m span, 4 lanes |

**Table 8.** Weight (in kg) of steel reinforcement for bending moment—longitudinal reinforcement. Bridge identifiers are explained in Table 7.

| Bridge Identifier | LFEA 0° | LFEA 15° | LFEA 30° | LFEA 45° | LFEA 60° | AASHTO LRFD 0° | AAHTO LRFD 60° |
|---|---|---|---|---|---|---|---|
| 1 | 659 | 824 | 659 | 495 | 264 | 824 | 659 |
| 2 | 1275 | 1275 | 1275 | 879 | 484 | 1275 | 1099 |
| 3 | 2033 | 2033 | 2033 | 1374 | 989 | 2033 | 1703 |
| 4 | 3231 | 3231 | 2835 | 2440 | 1649 | 3231 | 2440 |
| 5 | 1083 | 1354 | 1083 | 812 | 433 | 1354 | 1083 |
| 6 | 2094 | 2094 | 1806 | 1444 | 794 | 2094 | 1806 |
| 7 | 3340 | 3340 | 3340 | 2257 | 1354 | 3340 | 2799 |
| 8 | 4658 | 5308 | 4658 | 3358 | 1950 | 5308 | 4008 |
| 9 | 1507 | 1507 | 1507 | 1130 | 452 | 1884 | 1507 |
| 10 | 2914 | 2914 | 2512 | 2010 | 1306 | 2914 | 2512 |
| 11 | 4647 | 4647 | 3894 | 3140 | 1884 | 4647 | 3894 |
| 12 | 6481 | 7385 | 6481 | 4672 | 2713 | 7385 | 5577 |
| 13 | 1931 | 1931 | 1448 | 1062 | 579 | 2414 | 1931 |
| 14 | 3219 | 3219 | 2575 | 1931 | 1159 | 3733 | 3219 |
| 15 | 4989 | 5954 | 4989 | 4023 | 2414 | 5954 | 4989 |
| 16 | 8304 | 8304 | 7145 | 5986 | 3476 | 9462 | 7145 |

**Table 9.** Weight (in kg) of steel reinforcement for bending moment—transverse reinforcement. Bridge identifiers are explained in Table 7.

| Bridge Identifier | LFEA 0° | LFEA 15° | LFEA 30° | LFEA 45° | LFEA 60° | AASHTO LRFD 0° |
|---|---|---|---|---|---|---|
| 1 | 132 | 198 | 274 | 297 | 201 | 198 |
| 2 | 264 | 264 | 404 | 453 | 347 | 264 |
| 3 | 330 | 374 | 566 | 648 | 566 | 385 |
| 4 | 528 | 528 | 831 | 936 | 818 | 528 |
| 5 | 217 | 379 | 515 | 482 | 282 | 271 |
| 6 | 433 | 578 | 823 | 831 | 527 | 433 |
| 7 | 542 | 803 | 1201 | 1273 | 831 | 632 |
| 8 | 867 | 1094 | 1668 | 1842 | 1257 | 867 |
| 9 | 301 | 512 | 678 | 603 | 347 | 377 |
| 10 | 603 | 834 | 1156 | 1075 | 663 | 603 |
| 11 | 754 | 1218 | 1758 | 1670 | 1042 | 879 |
| 12 | 1206 | 1733 | 2547 | 2592 | 1658 | 1206 |
| 13 | 386 | 570 | 743 | 647 | 386 | 483 |
| 14 | 772 | 978 | 1313 | 1172 | 760 | 772 |
| 15 | 966 | 1513 | 2108 | 1915 | 1223 | 1126 |
| 16 | 1545 | 2221 | 3148 | 3013 | 1989 | 1545 |

**Table 10.** Difference (in kg) between total weight of reinforcement determined with LFEA and AASHTO LRFD. The difference is determined as the amount of steel designed using LFEA minus the amount required by AASHTO LRFD. Bridge identifiers are explained in Table 7.

| Bridge Identifier | 0° | 15° | 30° | 45° | 60° |
|---|---|---|---|---|---|
| 1 | −231 | 0 | −89 | −231 | −392 |
| 2 | 0 | 0 | 141 | −207 | −532 |
| 3 | −55 | −11 | 181 | −396 | −533 |
| 4 | 0 | 0 | −92 | −382 | −501 |
| 5 | −325 | 108 | −27 | −60 | −639 |
| 6 | 0 | 144 | 101 | −253 | −917 |
| 7 | −90 | 172 | 569 | −442 | −1246 |
| 8 | −650 | 227 | 152 | −975 | −1668 |
| 9 | −452 | −241 | −75 | −151 | −1085 |
| 10 | 0 | 231 | 151 | −30 | −1145 |
| 11 | −126 | 339 | 126 | −716 | −1846 |
| 12 | −904 | 528 | 437 | −1326 | −2412 |
| 13 | −579 | −396 | −705 | −705 | −1448 |
| 14 | −515 | −309 | −618 | −888 | −2073 |
| 15 | −1126 | 386 | 16 | −177 | −2478 |
| 16 | −1159 | −483 | −715 | −2008 | −3225 |

Figure 12a and Table 8 present the comparison for the longitudinal reinforcement. For skew angles up to 15°, the difference in weight between the two analysis methods is about 350 and 150 kg for 0° and 15°, respectively. Increasing skew angles result in larger differences: 600, 1100, and 1500 kg for 30°, 45°, and 60°, respectively. The AASHTO LRFD approach resulted in larger amounts of longitudinal steel, with only a few cases equal to LFEA. These cases corresponded to bridges with straight geometries or with a skew angle of 15° (see Figure 7). Figure 12b and Table 9 present the comparison for transverse reinforcement. For straight bridges, LFEA results in about 50 kg less of steel. As the skew increases, the reinforcement design with AASHTO LRFD is insufficient to carry the transverse bending moments. The difference with LFEA is in the order of 200, 550, 550, and 150 kg for 15°, 30°, 45°, and 60°, respectively.

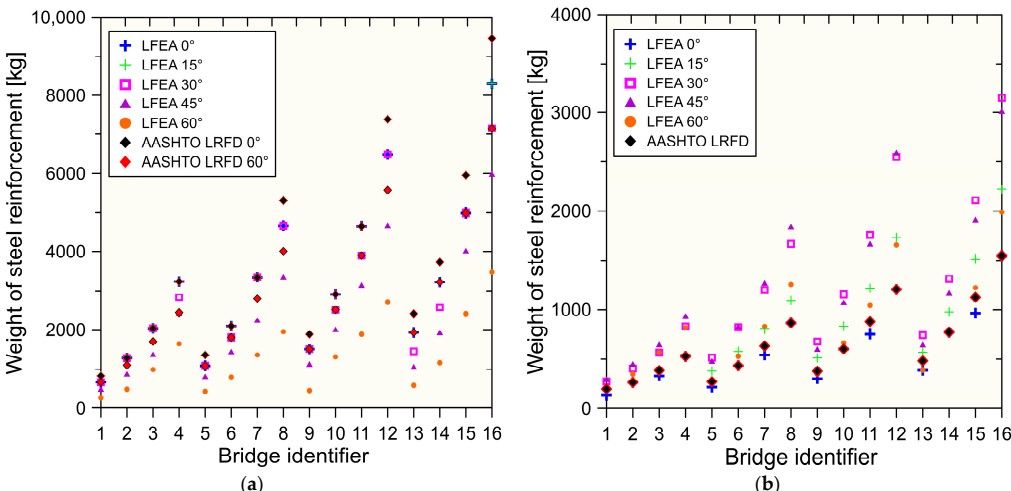

**Figure 12.** Weight (in kg) of steel reinforcement for bending moment. (**a**) Longitudinal reinforcement, and (**b**) transverse reinforcement, with bridge identifiers explained in Table 7.

Figure 13 and Table 10 show the total amount of flexural steel (sum of longitudinal and transverse steel). This comparison allows us to determine the difference between the two analysis approaches in terms of total steel weight. Positive values in Figure 13 and Table 10 indicate a lower total steel weight required by AASHTO LRFD than when using LFEA. For most of the 15° and 30° case study bridges, the total steel weight designed with AASHTO LRFD was slightly lower than when using LFEA. For the majority of the remaining skewed cases, the total steel weight designed with AASHTO LRFD was much higher than when using LFEA. As the skew angle increases, AASHTO LRFD overestimates the longitudinal bending moments but underestimates the required steel for transverse moments since the AASHTO procedures do not consider the effect of skew on the transverse reinforcement. For the 15° and 30° skew angles, the differences between the two analysis methods nearly balance out. For the 45° and 60° skew angles, the overestimation of main longitudinal bending moments per AASHTO LRFD simplified procedures exceeds the underestimation of secondary transverse steel, leading to higher total amounts of steel.

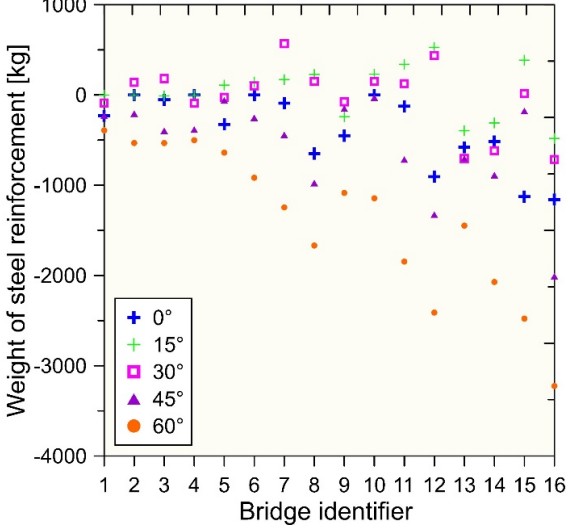

**Figure 13.** Difference (in kg) between total weight of reinforcement determined with LFEA and AASHTO LRFD. The difference is determined as the amount of steel designed using LFEA minus the amount required by AASHTO LRFD. Bridge identifiers are explained in Table 7.

## 5. Discussion

Slab bridges are widely used and form an important element in our infrastructure. Even though straight geometries are preferable, urban or geographical constraints may make the selection of skewed geometries necessary. This paper presents the results of a parametric study to determine the applicability of AASHTO LRFD for simply supported reinforced concrete skewed slab bridges. Ninety case study bridges with different span lengths, skew angles, number of lanes, and material properties were used in this study. The bridges were analyzed with the AASHTO LRFD simplified procedures using hand calculations, and with LFEA using SCIA Engineer 20 [38]. In a final step, the required longitudinal, transverse, and shear reinforcement were designed according to the AASHTO LRFD for both analysis methods.

Our observed reductions in maximum longitudinal bending moments for skewed geometries are aligned with the parametric study conducted by Menassa. Menassa found a 50% bending moment reduction for the 50° case study bridges [30]. For the bridges analyzed in our parameter study, a 45% and 65% bending moment reduction was obtained for the 45° and 60° cases, respectively. A main difference between our study and the work by Menassa lies in the magnitude of the bending moments for the exterior strips calculated with AASHTO LRFD. In Menassa's work, these moments were smaller than those of the interior strips [30]. Contrarily, in our work, all exterior strip bending moments were higher than those in the interior strip. This difference is explained by the fact that Menassa did not consider concrete barrier loading. We considered concrete barrier loading and assumed that it acted solely on the exterior strips when using AASHTO LRFD, as recommended by Rodríguez [36].

The moment reduction coefficients developed by Théoret are not directly comparable to the moment reductions observed in this study because Théoret's values were computed for the relation of width over length [9]. We can observe, however, that both Théoret's work and ours indicate a significant reduction of the longitudinal bending moment reduction as the skew angle increases.

Based on the width factor determined with LFEA, we can conclude that the assumption of the barrier acting solely on the exterior strips can be overly conservative for exterior strip design when the simplified procedures are used. The results from Section 4.2 show the large distribution capacity for bending moments of concrete slab bridges. The reader should note here that this distribution capacity is based on linear elastic calculations. In reality, after cracking, the distribution capacity of these bridges is even larger. In consequence, we recommend distributing the weight of the barrier over the entire slab width when using simplified code procedures.

A common trend is observed for the transverse bending moments in the parametric studies from Théoret and Menassa, which used more refined analysis than AASHTO LRFD. While these moments are practically negligible for straight geometries, they increase significantly as the skew increases [9,30]. In our study, we also found that the transverse bending moments are negligible for straight geometries. The magnitude of these moments increased from 0° up to 30° or 45°, depending on the width and span length. We observed a significant decrease at 60°. This difference in trend can be explained by how the width of the bridge is accounted for in the studies. Both Théoret and Menassa keep the overall width constant as the skew increases. We considered the width to be dependent on the skew as it is related to the actual lane layout and driving direction. In consequence, when the skew increased, the trajectories of the principal stresses shifted towards the secondary direction. However, they did not concentrate as much because the larger width allows for more distribution.

For shear design, there is no clear agreement [32]. This study followed the design provisions from AASHTO LRFD based on the Modified Compression Field Theory [37], and applied the recommendations from Lipari [32] and Lantsoght [42]. We extended the proposals from Lipari [32] to quantify the influence of his recommendations on the shear reinforcement in simply supported reinforced concrete slabs. None of the 80 case study

bridges analyzed require shear reinforcement. The LFEA procedure can capture the stress concentrations close to the critical shear section at the obtuse corner in skewed bridges. The design procedure from Lipari was also validated [32] for all case study bridges. Our approach can be used in practice for the design of simply supported reinforced concrete skewed slab bridges.

Our parametric study aimed at comparing AASHTO LRFD simplified procedures and LFEA in a practical way. To quantify the differences between both approaches, we analyzed the weight of steel resulting from each design. Such a practical comparison is not provided in the parameter studies from the literature. In general, using LFEA allows us to reduce the amount of longitudinal steel while fulfilling the transverse bending moment demand in skewed slab bridges.

We identify a few topics of future research. The first topic to explore further is the influence of the width. We used a practical lane layout to determine the width and defined the width in terms of driving direction. Our parameter studies can be extended by looking at geometries in which the width is defined parallel to the support line so that the support line width remains constant. The results from this study suggest that transverse moments as well as shear forces would further increase with this modification. A second topic for future research is the critical position of the truck for shear. We used a position at 600 mm from the barrier, as suggested in AASHTO LRFD. However, concentrated loads closer to the edge result in larger stress concentrations in the obtuse corner. Therefore, it would be interesting to verify the shear demand with a more critical truck positioning. Third and most important, experimental research on skewed slab bridges is very limited. Therefore, we recommend conducting experiments on skewed slabs to better understand the behavior at the ultimate limit state. Such studies can validate if LFEA is adequate for skewed slab bridge analysis or if a more refined analysis would be justified. The experimental results can be used to verify the assumption of the shear distribution width of 4 *d*, which was derived for straight slabs.

## 6. Conclusions

In this article, the results from a parametric study on the applicability of AASHTO LRFD for the design of simply supported, reinforced concrete skewed slab bridges were presented. We drew the following conclusions:

- AASHTO LRFD simplified procedures are unable to accurately capture the reduction in magnitude of longitudinal bending moments as the skew angle increases. AASHTO LRFD results are only comparable to LFEA for skew angles up to 15°.
- Slab bridges have a large distribution capacity for the main longitudinal bending moment. This suggests that the same main longitudinal bending moment reinforcement can be provided for the entire width of the bridge.
- For AASHTO LRFD hand calculations, we recommend distributing the barrier weight over the entire width instead of over the exterior strip only.
- Distribution reinforcement per AASHTO LRFD cannot be used for the design of skewed slab bridges. As the skew increases, additional transverse reinforcement has to be provided to meet the moment demands.
- AASHTO LRFD simplified procedures do not capture skew effects for shear design. Thereby, Lipari's [32] and Lantsoght's [42] suggestions are recommended for use in practice with the AASHTO LRFD design provisions for skewed reinforced concrete slab bridges.
- Using AASHTO LRFD instead of LFEA for obtaining design longitudinal bending moments is conservative for skewed reinforced concrete slab bridges. However, using AASHTO LRFD instead of LFEA for obtaining design transverse bending moments and shears can be unconservative or unsafe.
- Using LFEA for analysis instead of AASHTO LRFD simplified procedures generally leads to a reduction in total reinforcement steel weight and can thus be considered cost-effective. As such, we recommend the use of LFEA for the design of skewed slab bridges.

**Supplementary Materials:** The following are available online at https://doi.org/10.5281/zenodo.47 41292, PDF document S1: Calculation Memories, PDF document S1: Summary of Results, Zip folder S1: SCIA Engineering Models. Calculation Memories is a PDF document that contains a compilation of the detailed calculations performed for the analysis and design of the reinforced concrete solid slab bridges. This document is divided into two main sections: Bridge Designs Performed through Hand Calculations and Bridge Designs Performed with SCIA Engineer Software. The calculations are executed for the straight bridges as well as for the 15°, 30°, 45°, and 60° skew angles. Summary of Results is a PDF document that contains a compilation of the summary of results for the analysis and design of the reinforced concrete solid slab bridges. This document is divided into two main sections: Bridge Designs Performed through Hand Calculations and Bridge Designs Performed with SCIA Engineer Software. The results are summarized for the straight bridges as well as for the 15°, 30°, 45°, and 60° skew angles. SCIA Engineering Models is a zip folder with all 80 models performed in SCIA Engineer.

**Author Contributions:** Conceptualization, E.O.L.L. and L.M.; methodology, L.M.; software, L.M.; validation, E.O.L.L.; formal analysis, L.M.; investigation, L.M.; resources, L.M.; data curation, L.M.; writing—original draft preparation, L.M.; writing—review and editing, E.O.L.L.; visualization, L.M.; supervision, E.O.L.L. All authors have read and agreed to the published version of the manuscript.

**Funding:** This research received no external funding.

**Data Availability Statement:** All calculation memories, results, and bridge models from SCIA Engineer are available through the open-access repository Zenodo at https://doi.org/10.5281/zenodo.4741292 (accessed on 6 May 2021).

**Conflicts of Interest:** The authors declare no conflict of interest.

**Notation:**

| | |
|---|---|
| $A_{s,calc}$ | Area of steel calculated from flexural design |
| $E$ | Equivalent width for interior and exterior strips |
| $L$ | Span length taken as the dimension of the free edge |
| $L_1$ | Modified span length |
| $M_{l,max}$ | Maximum main longitudinal bending moment |
| $M_{t,max}$ | Maximum secondary transverse bending moment |
| $N_L$ | Number of lanes |
| UC | Unity check defined as the ratio of ultimate shear demand to concrete shear capacity |
| $V_u$ | Ultimate shear demand |
| $W$ | Width taken as the dimension perpendicular to the free edge |
| $W_1$ | Modified width |
| $W_{eff}$ | Effective width for main longitudinal bending moment |
| $W_{fact}$ | Width factor defined as the ratio of effective width for main longitudinal bending moment to width of the bridge |
| $X$ | Direction of the *x*-axis |
| $Y$ | Direction of the *y*-axis |
| $b$ | Full width of a section cut |
| $b_{eff}$ | Effective width of a section cut |
| $d$ | Effective depth to the main longitudinal reinforcement |
| $d_v$ | Effective shear depth |
| $f(x)$ | Function describing the unitary bending moment along a section cut |
| $f_{max}$ | Peak unitary bending moment or shear force within a section cut |
| $m$ | Multiple presence factor |
| $m_o$ | Magnitude of bending moment in the direction of the principal shear force |
| $m_x$ | Magnitude of bending moment in the x direction |
| $m_{xy}$ | Magnitude of torsion effects along the x direction |
| $m_y$ | Magnitude of bending moment in the y direction |
| $r$ | Reduction factor for main longitudinal bending moments |
| $t$ | Thickness of the slab in millimeters |
| $v_o$ | Magnitude of principal shear force |

| | |
|---|---|
| $v_x$ | Magnitude of shear force in the x direction |
| $v_y$ | Magnitude of shear force in the y direction |
| $\beta$ | Resistance parameter that shows the ability to transmit both tension and shear from diagonally cracked concrete [5] |
| $\alpha$ | Skew angle in degrees formed between the *y*-axis and the free edge |
| $\alpha_x$ | Skew angle in degrees formed between the *x*-axis and the free edge |
| $\varphi$ | Bar diameter dimension |
| $\theta_o$ | Angle between the *x*-axis and the direction of the principal shear force |
| $\phi V_c$ | Concrete shear capacity |

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
