# Peer review of "Parametric Study on the Applicability of AASHTO LRFD for Simply Supported Reinforced Concrete Skewed Slab Bridges"

_infrastructures, doi:10.3390/infrastructures6060088_

Round 1

Reviewer 1 Report

This paper highlights the parametric study on the applicability of AASHTO LRFD for simply supported reinforced concrete skewed slab bridges. The paper cannot be accepted in the present form as it needs further improvements. First of all, the article must be proofread in detail. There are numerous typos.

  1. Abstract: The text must be carefully revised. Some sentences contain mistakes. In a research paper, it is expected that the introduction section briefly explains the starting background and, even more important, the originality (novelty) and relevancy of the study is well established. Once this is done, the hypothesis and objectives of the study need to be addressed, as well as a brief justification of the conducted methodology.
  • Line 10: Define AASHTO LRFD at least once in the manuscript.
  • Line 37: Recheck once again.
  • Underscore the scientific value-added to your paper in your abstract.
  • Clearly discuss what the previous studies that you are referring to are. What are the Research Gaps/Contributions?
  1. The introduction part does not have a flow or direction. It has too many different medical terminologies thrown randomly. Proper references need to be used rather than using others. Language can be improved. The sentences are half-constructed or incomplete so that the readers are expected to fend for themselves to understand their meaning.
  • Line 53: Consider rewriting the sentence to remove the unclear reference.
  • Line 63: It seems that there is a pronoun problem here.
  • Line 74-75: Your sentence may be unclear or hard to follow. Consider rephrasing.
  • Line 105: It appears that that might occur may be unnecessary in this sentence. Consider removing it.
  • It seems that you have an unnecessary comma throughout the manuscript. Consider removing the comma.
  • Figures 1-5: are of very poor quality. Replace it.
  • Line 206-207: Your sentence may be unclear or hard to follow. Consider rephrasing.
  • Line 338-339: Your sentence may be unclear or hard to follow. Consider rephrasing.
  • Line 409-412: Your sentence may be unclear or hard to follow. Consider rephrasing.
  • Line 423-425: Your sentence may be unclear or hard to follow. Consider rephrasing.
  1. Author must be enriching the references with the latest developments in the field. Some of the recent references can be added. The authors have not paid attention to previous research papers and concerns.
  2. The innovation contribution of this article is not clearly stated. The research contributions should be highlighted in the revised manuscript. There is a certain lack of a clear line and message, and my strong advice to the authors would be to consider the overall structure and to either significantly shorten the manuscript.
  3. Conclusions should be more concrete. Try to introduce a new section. They should be summarized in 3-5 bullet points that clearly show the conclusions of this study. In addition, since it is a review, it is essential to indicate future lines of research. What would be the ways, in the real world, to change/improve the observed situation.

The list could go on, but the bottom line is that the authors need to rewrite the paper or even reconsider the research content before it could be considered for publication in this journal. I would recommend ‘Major revision’ for this paper.

Reviewer 2 Report

The article is suitable for publication

Author Response

We would like to thank the reviewer for the report. 

Reviewer 3 Report

Dear Authors,

Thank you for your manuscript, it is definitely interesting study but written in a bit hectic way that clear engineering approach is not seen (please skip long sentences, make them shorter and more clear), please find following comments enclosed:

  1. Abstract has to be shortened and rewritten with a clear structure (shorter sentences for a better readability): WHY, problem, novelty, solution and results.

Lines 34-107: Please rewrite and make it more clear structured (try to avoid to much wording talking about the same thing). Bring more references to the actual work and not to the theory written in the manuals. Underline the novelty of your paper and study. In the last paragraph identify what is your paper about, what is so unique in it to be published.

Figure 2 and other figures, please make sure you have large enough font for a better readability, currently it is not sufficient.

Where the 80 case study bridges come from? Are they classified in categories? What is the difference between them? Can you please bring a table or flow chart for a better visualization?

Line 295: indeed bring a table to show which cases have skew angle greater than 0°, bring your description per category, compare categories.

Figure 6, 9 etc explain all your charts one by one, indicated what is the difference between them. Please don’t bring a general description to your figures. Even thought have numbers in table, there is still has to be description. Bring conclusion to the set of charts in figure.

There is no need to add units to all numbers in columns, once you have mentioned it in the header.

The conclusions are way too long and it would be much better to have a short bullet points to what is done and what is novel. The part of your conclusions you bring to discussion part.

Round 2

Reviewer 1 Report

The authors have incorporated all the changes in the revised manuscript. It can be accept in present form.

Reviewer 3 Report

Dear Authors, the manuscript was very well elaborated and there are no further comments.